# CaSBRE: Causality-inspired Semi-supervised Biomedical Relation Extraction

## Abstract

Biomedical interaction relations, such as chemical-protein interactions (CPIs) and gene-disease associations (GDAs), are crucial for advancing drug discovery and clinical treatments. However, the vast diversity of biomedical entities and the limited availability of labeled data pose significant challenges to accurately modeling these interactions using traditional supervised learning approaches. These methods often overfit to spurious feature-label correlations in the scarce labeled relations, leading to poor generalization to unseen biomedical entities. To overcome these challenges, we introduce CaSBRE, a causality-inspired semi-supervised learning framework designed to disentangle and mitigate the impact of such spurious correlations. CaSBRE includes two core components: (i) Feature Disentanglement, which separates causal from spurious features by identifying and exploiting discrepancies between their correlations in labeled and unlabeled data; and (ii) Do-calculus Interaction Inference, which marginalizes the influence of spurious features on relation predictions. Through extensive experiments on CPI and GDA tasks, we demonstrate that CaSBRE substantially outperforms state-of-the-art methods, particularly in generalizing to previously unseen biomedical entities, thereby providing a robust and scalable solution for biomedical relation extraction.

## 1 Introduction

Biomedical relations, such as chemical-protein interactions (CPIs), gene-disease associations (GDAs), and protein-protein interactions (PPIs), play a crucial role in various biomedical studies, including drug discovery Kim et al. (2021) and clinical treatments Grant et al. (2018). However, given the vast number of biomedical entities spanning multiple domains—such as chemical molecules, proteins, and diseases—it is extremely costly to fully annotate the relationships between these entities based on wet experimental results or expert knowledge. Consequently, there has been a growing interest in leveraging computational approaches, particularly those based on machine learning, to automate the discovery of these relationships. Most current methods rely on supervised machine learning, which presupposes the availability of large-scale annotated data. However, this is not always feasible, particularly for new biomedical entities, like newly-developed drug molecules. To overcome this challenge, semi-supervised relation extraction methods, which use a small amount of annotated data along with a large set of unlabeled instances, have shown increasing promise Zhang & Lu (2019); Wen et al. (2023).

Semi-supervised learning (SSL) has been widely explored across various domains and can be broadly classified into three paradigms based on how unlabeled data is utilized: *(i) consistency regularization* Rasmus et al. (2015), *(ii) self-labeling* Sohn et al. (2020); Tai et al. (2021), and *(iii) generative models* Kingma et al. (2014); Odena (2016). Despite its success in various domains, particularly computer vision, it faces significant challenges in the realm of biomedical relation extraction. First, **data augmentation infeasibility**: in biomedical relation extraction, unlike in computer vision, introducing noise or perturbations to data—such as molecular substructures or protein embedding vectors—can significantly alter their biomedical properties, making traditional augmentation techniques impractical. Second, unlike multi-class classification tasks, biomedical relation extraction commonly involves **binary interactions** which limits the use of category dependencies in pseudo-labeling strategies and the modeling of joint feature-label distributions in generative models, such as semi-supervised variational autoencoders Kingma et al. (2014). Additionally, **paired data structure**: unlike image

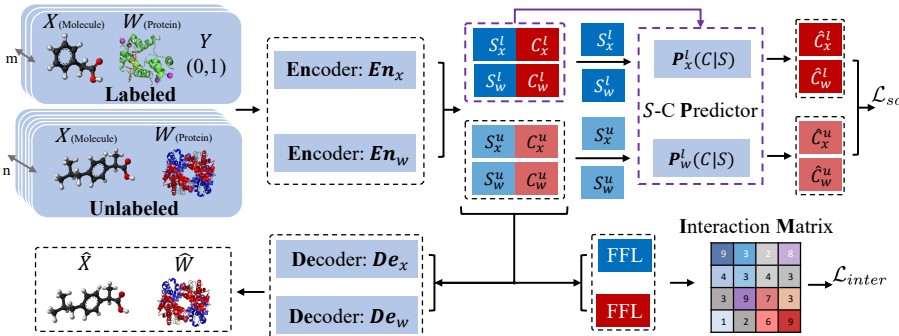

Figure 1: **Overview.** We disentangle the spurious features $S$ from causal ones $C$ under the guidance of $S$-$C$ predictors which are trained to detect the discrepancy of $S$-$C$ correlations between labeled training data and unlabeled data.

data that comes from a single domain, biomedical relations involve paired data from two distinct domains, meaning the interactions cannot be determined based on inputs from just one domain.

The aim of SSL is to utilize large amounts of unlabeled data to mitigate overfitting on a limited set of training annotations and effectively capture generalizable feature-label relationships. In the biomedical domain, the challenges of SSL are further compounded by the presence of spurious feature-label correlations, which stem from the following two primary factors. **(i) Biased relation annotations** Cheung et al. (2013); Haynes et al. (2018): currently reported relations are biased for various reasons including historical and research focus bias and biological bias. For instance, in CPIs, well-studied proteins, such as GPCRs or kinases, are more likely to be reported. Similarly, in GDAs, common complex diseases such as cancer or cardiovascular diseases tend to have more reported gene-disease pairs compared to rare monogenic diseases. **(ii) Scarcity of relation annotations** Zhang et al. (2024): the currently available biomedical relations shrink compared to the vast potential combinations across millions of biomedical entities. Furthermore, the pairwise nature of these interaction relations results in a quadratic increase in spurious effects, significantly hindering the generalization capabilities of current SSL models in biomedical relation extraction.

In this paper, we propose Causality-inspired Semi-supervised Biomedical Relation Extraction (CaS-BRE), which reduces the spurious feature-label correlations. CaSBRE consists of two key components: **(i) feature disentanglement** aims to separate the learned features into causal part $C$ and spurious part $S$. The strategy hinges on the assumption that while strong correlations between $S$ and $C$ may exist in the limited labeled data (characterized by a conditional distribution $P^l(C|S)$), these correlations may not generalize to large-scale unlabeled data (where the corresponding distribution is $P^u(C|S)$), namely $P^l(C|S) \neq P^u(C|S)$. To achieve this, we train an $S$-$C$ predictor to approximate $P^l(C|S)$, and guide the feature disentanglement by maximizing the discrepancy between $P^l(C|S)$ and $P^u(C|S)$. **(ii)** Building on these disentangled features, we perform relation inference using a causality-inspired **do-calculus relation inference** strategy to neutralize the influence of spurious features $S$. Similar to the backdoor criterion Pearl (2009), it marginalizes out the effects of $S$ by averaging predictions over the prior distribution of $S$, rather than its observed conditional distribution. This prevents spurious features learned during training from skewing the predictions, thereby fostering more robust and generalizable relation extraction. Our primary contributions are summarized as follows :

- The introduction of a benchmark dataset encompassing chemical-protein interactions and gene-disease associations for a comprehensive evaluation of biomedical relation learning.

- The proposal of the first general semi-supervised framework for causal interaction learning, including an algorithm to jointly train the framework's causal interaction representation learning module and causal interaction inference.

- A thorough evaluation of state-of-the-art supervised and semi-supervised approaches on the benchmark dataset, demonstrating that our method significantly improves generalization to novel biomedical entities compared to existing models.

## 2 RELATED WORK

### 2.1 BIOMEDICAL RELATION LEARNING

Biomedical relations encompass a vast array of entities across multiple domains, involving tens of thousands of interactions. Common types of interactions include chemical-protein interactions (CPIs) Kim et al. (2021), drug-drug interactions (DDIs) Ryu et al. (2018), protein-protein interactions (PPIs) Hu et al. (2021), gene-disease associations (GDAs) Wu et al. (2019), and chemical-disease associations (CDAs) Zhou et al. (2020). These interactions can occur within the same data domain, such as DDIs and PPIs, or between different domains, like CPIs and GDAs. Each type of interaction plays a critical role in various downstream applications, such as drug development and disease treatment.

There are five primary approaches to modeling these interactions: (a) Early research explored shallow machine learning models, such as Adaboost and SVMs, which are less effective for interaction learning. These models require keeping entities from one domain fixed while predicting related entities from another domain, leading to suboptimal performance Liu et al. (2012). (b) Another approach involves formulating relation learning as link prediction on large-scale graphs. However, this method struggles to generalize to novel entities, such as novel diseases or chemical molecules Zitnik et al. (2018); Chandak et al. (2023). (c) Multi-layer perceptron networks (MLPs) have been introduced to learn these relations Ryu et al. (2018); Wen et al. (2023). (d) Convolutional neural networks (CNNs) have also been employed to model feature interactions between both domains Huang et al. (2021). (e) Additionally, by treating the problem as a user-item relation, recommendation system techniques have been explored for relation extraction Ye et al. (2021).

### 2.2 SEMI-SUPERVISED LEARNING

The effectiveness of semi-supervised learning hinges on the ability to utilize large amounts of unlabeled data, thereby reducing the risk of overfitting to the labeled data. There are generally three main approaches: (i) Representation consistency regularization, where consistency is enforced between the original input and its augmented versions (e.g., through image rotations) or within the model itself (e.g., using dropout), to enhance model generalization Rasmus et al. (2015); Wei & Gan (2023). (ii) The self-labeling method, which iteratively selects confident predictions on unlabeled data as pseudo labels for retraining, gradually improving the model's performance Sohn et al. (2020); Tai et al. (2021); Xiao et al. (2024). (iii) Generative models, such as variational autoencoders (VAE) and generative adversarial networks (GAN), are used to learn low-dimensional representations or to capture the joint distribution between representations and labels Kingma et al. (2014); Springenberg (2015); Odena (2016); Zhao et al. (2020b); Ren et al. (2023). These techniques mainly focus on exploiting the intrinsic properties of input data, such as images or inter-category dependencies, which may not be optimal for binary interaction learning. Additionally, they do not address the potential issue of spurious feature-label correlations within the small labeled dataset, which can undermine the model's generalization ability.

Semi-supervised bio-medical relation extraction is mostly investigated independently for each specific task. For example, a GAN model is explored to learn protein and chemical representations separately for chemical-protein interactions Zhao et al. (2020a). A multi-task semi-supervised learning method is built on auto-encoders to learn drug-drug interactionsChu et al. (2019). There are a few works such as Zhang & Lu (2019), which are built on VAE considering multiple interaction learning tasks simultaneously. Most of them are investigated under a standard scenario, which randomly selects labeled entity pairs for training/test, and does not consider the generalization to novel entities. It is challenging to predict the interactions involved with novel entities but also important for many related studies such as novel drug development or novel disease treatment.

## 3 THE PROPOSED METHOD

### 3.1 PROBLEM SETUP

We study the task of learning a relation inference model to map an input pair $(X, W)$ onto a label space $y \in \mathcal{Y}$, where $y$ ranges from 0 to 1. In this context, $X \in \mathcal{X} \subseteq \mathbb{R}^{d_x}$ and $W \in \mathcal{W} \subseteq \mathbb{R}^{d_w}$ represent

two different data domains. For example, in CPI data, chemicals ($X$) and proteins ($W$) represent two distinct domains. Given a training dataset containing $m$ labeled $(x, w)$ pairs, $\mathcal{D}^l = \left\{ (x_i^l, w_i^l, y_i^l) \right\}_{i=1}^m$ which covers $m_x$ and $m_w$ entities of $X$ and $W$, respectively, and $n_x$ and $n_w$ unlabeled input instances from $X$ and $W$, denoted as, $\mathcal{D}_x^u = \{(x_i)\}_{i=1}^{n_x}$ and $\mathcal{D}_w^u = \{(w_i)\}_{i=1}^{n_w}$, with $m_x \ll n_x$ and $m_w \ll n_w$. The goal of SSL is to train a model $h(x, w; \theta): (\mathbb{R}^{d_x}, \mathbb{R}^{d_w}) \to [0, 1]$ using this combination of labeled and unlabeled data. Here the $h(x, w; \theta)$ is parameterized by $\theta \in \Theta$.

We define the features that are causal for the relation interactions as causal features ($C$) and those that do not contribute to interactions but are highly correlated with the causal features as spurious features ($S$). Following previous work about the bias and scarcity of biomedical relation annotations Cheung et al. (2013); Haynes et al. (2018); Zhang et al. (2024), we assume that the correlations between $S$ and $C$ in labeled data may not generalize to unlabeled data in biomedical relation extraction. Therefore, in labeled training data, the strong correlations between $S$ and $C$ may lead an interaction model to mistakenly prioritize $S$ as important features, resulting in poor generalization to unseen data. Our goal is to disentangle the learned representation into causal and spurious components, thereby steering the interaction learning process to focus on the causal features.

## 3.2 Overview

Given an input pair $(X, W)$, the CaSBRE computation process involves three main components, as illustrated in Figure 1. **(i)** CaSBRE first disentangles the learned representations into causal and spurious parts using the Feature Disentanglement Network (FDN), which is built on variational autoencoders Doersch (2016). The FDN consists of two encoders, $\mathbf{En}_x$ and $\mathbf{En}_w$, which encode the input data into low-dimensional representations. These representations are subsequently used by two decoders, $\mathbf{De}_x$ and $\mathbf{De}_w$, to reconstruct the original inputs. Additionally, two $S$-$C$ predictors, $\mathbf{P}_x$ and $\mathbf{P}_w$, are trained to capture the correlations between spurious features (S) and causal features (C) existing in the labeled training data. **(ii)** The Duplex Interaction Module (DIM), comprising feedforward layers (FFL) and an interaction matrix $\mathbf{M}$, is introduced to capture fine-grained feature interactions between the two domains. **(iii)** Finally, do-calculus interaction inference is performed to minimize the prediction's reliance on spurious features.

## 3.3 Feature Disentanglement Network

For domains $X$ and $W$, the FDN disentangles the learned features into causal and spurious parts, represented as $(C_x, S_x)$ and $(C_w, S_w)$. To achieve this, variational auto-encoders are employed to encode the inputs into low-dimensional representations, which are then guided to distinguish between $S$ and $C$ using two $S$-$C$ predictors. Since the operations for the $X$ and $W$ domains are similar in subsequent steps, we will refer to either domain as $X$ and omit the subscript $x$ for simplicity unless otherwise specified.

### 3.3.1 Representation Learning

We utilize variational auto-encoders to learn low-dimensional representations of the inputs, trained to approximate the underlying data generative process using large-scale unlabeled data. For domain $X$, an encoder network $\mathbf{En}_x$ maps the inputs $X$ into low-dimensional representations, $Z = [C, S]$, where $[\cdot, \cdot]$ means concatenation. Correspondingly, a decoder network $\mathbf{De}_x$ attempts to reconstruct the original inputs from these representations. The encoder and decoder are jointly optimized to minimize the reconstruction error $\mathcal{L}_{con}$, defined as,

$$\mathcal{L}_{con}^X = \left\| \mathbf{De}_x\left( \mathbf{En}_x(X) \right) - X \right\|^2, \tag{1}$$

The learned representation is guided to match prior Gaussian distributions $P(C)$ and $P(S)$ by the objective $\mathcal{L}_{kl}$,

$$\mathcal{L}_{kl}^X = \mathbb{KL}\left( Q(C|X, \mathbf{En}_x), P(C) \right) + \mathbb{KL}\left( Q(S|X, \mathbf{En}_x), P(S) \right) \tag{2}$$

where $\mathbb{KL}(\cdot, \cdot)$ denotes the Kullback-Leibler divergence, and representation distributions $Q(C|X, \mathbf{En}_x)$ and $Q(S|X, \mathbf{En}_x)$ are parameterized by the encoder $\mathbf{En}_x$. The prior distribution $P(C) = P(S)$ is modeled as a standard Gaussian distribution $\mathbb{N}(0, 1)$.

### 3.3.2 $S$-$C$ DISENTANGLEMENT

We model the strong $S$-$C$ correlation in the labeled data through an $S$-$C$ predictor $\mathbf{P}_x = P^l(C|S)$ trained on labeled training data. The $S$-$C$ predictor approximates the $S$-$C$ correlation by predicting $C$ from $S$ and minimizing the prediction errors between the predicted $C$ and the actual $C$ obtained by $\mathbf{En}_x$, using $\mathcal{L}_c^X = ||\mathbf{P}_x(S^l) - C^l||_2$ trained on only labeled data.

To guide encoders to distinguish the spurious features $S$ from causal ones $C$, we further introduce $\mathcal{L}_{sc}$ utilizing both labeled and unlabeled data. Since we assume the $S$-$C$ correlations in the labeled data may not generalize to unlabeled data, i.e., $P^l(C|S) \neq P^u(C|S)$, the $S$-$C$ predictors would be worse in approximating the $S$-$C$ correlations in unlabeled data than in labeled data, namely having much larger prediction errors. We denote the prediction error of $C$ in labeled data as $\mathrm{Err}_{sc}^l = ||C^l - \mathbf{P}_x(S^l)||^2$ and in unlabeled data as $\mathrm{Err}_{sc}^u = ||C^u - \mathbf{P}_x(S^u)||^2$. We guide the feature disentanglement by maximizing the error discrepancy of the $C$ predictions between the labeled data and the unlabeled data:

$$\mathcal{L}_{sc}^X = \max\{\mathrm{Err}_{sc}^l - \mathrm{Err}_{sc}^u + t_x, 0\}, \tag{3}$$

where $t_x$ is the threshold that regulates the discrepancies of $S$-$C$ conditional distributions between the labeled and unlabeled data. $\mathcal{L}_{sc}^X$ explicitly guides feature disentanglement by optimizing the encoders to produce $S$ and $C$ with differing dependencies between labeled and unlabeled data. We optimize $\mathcal{L}_c$ and $\mathcal{L}_{sc}$ in an interactive manner (see Section 3.5).

## 3.4 CAUSAL INTERACTION INFERENCE

### 3.4.1 DUPLEX INTERACTION MODULE

We aim to model fine-grained feature interactions between two domains through the duplex interaction module (DIM), which consists of feedforward layers (FFL) and an interaction matrix $\mathbf{M}$. The FFL further transforms the latent representations obtained by the $\mathbf{En}$ to bridge the gap between VAE and interaction learning as the VAE-learned representation might not be directly optimal for the linear interaction modeling by the interaction matrix. Thus, interaction prediction for the paired $(X, W)$ is: $\hat{Y} = \mathrm{FFL}(\mathbf{En}_x(X)) \times \mathbf{M} \cdot \mathrm{FFL}(\mathbf{En}_w(W))^T$.

### 3.4.2 RELATION INFERENCE VIA DO-CALCULUS

Once the FDN has disentangled the causal and spurious features during training, our subsequent goal is to ensure that during inference, the final relation predictions are not dependent on these potentially misleading spurious features. To achieve this, we employ a do-calculus inference strategy following the backdoor criterion in do-calculus Pearl (2009)). Specifically, we marginalize the spurious features from the *prior* distributions to obtain their average effects on the relation prediction as shown in Algorithm 2:

$$Y_{\mathrm{pred}} = P[Y|(X, W)] = \sum_{k_x=1}^{K} \sum_{k_w=1}^{K} \mathrm{FFL}([\mathbf{En}_x(X), S_x^{k_x}]) \times \mathbf{M} \cdot \mathrm{FFL}([\mathbf{En}_w(W), S_x^{k_w}]^T), \tag{4}$$

where $K$ denotes the sampling number of $S$ from the prior Gaussian distribution $\mathbb{N}(0, 1)$, and $S_x^{k_x}$ and $S_x^{k_w}$ denote the sampled $S$ for $X$ and $W$, respectively.

## 3.5 TRAINING AND INFERENCE OF CASBRE

The training process of CaSBRE is designed to ensure the effective disentanglement of spurious and causal features, guided by $S$-$C$ predictors, and prevent the relation predictions from depending on the spurious features. At each training stage of encoders, it is expected that the $S$-$C$ predictors already perfectly estimate the $S$-$C$ correlations. Therefore, the $S$-$C$ predictors and the other components, including the encoders, are optimized in an interactive manner. The $S$-$C$ predictors are trained using the loss function $\mathcal{L}_c = \mathcal{L}_c^X + \mathcal{L}_c^W$. The encoders are trained by the objective $\mathcal{L}_{enc}$. It consists of three parts: (i) pair interaction prediction objective, $\mathcal{L}_{inter} = -\left(Y \log(\hat{Y}) + (1 - Y) \log(1 - \hat{Y})\right)$, which is a standard binary cross entropy loss ; (ii) generative representation learning objective, $\mathcal{L}_{con}$

Table 1: **Results on the chemical-protein interaction predictions** with AUROC (AUPRC) reported. S: Supervised approach.

| Setting | Method | Random-pair | Single-cross | Double-cross |
|---|---|---|---|---|
| 20% | S:DeepDDI Ryu et al. (2018) | 0.861±0.005 (0.621±0.010) | 0.799±0.017 (0.506±0.048) | 0.682±0.029 (0.360±0.048) |
| | S:MolTrans Huang et al. (2021) | 0.864±0.008 (0.610±0.027) | 0.807±0.016 (0.517±0.037) | 0.682±0.032 (0.350±0.054) |
| | S:KGE_NFM Ye et al. (2021) | 0.858±0.103 (0.621±0.011) | 0.800±0.112 (0.520±0.054) | 0.662±0.021 (0.343±0.025) |
| | M1 model Kingma et al. (2014) | 0.870±0.003 (0.618±0.020) | 0.811±0.013 (0.539±0.043) | 0.674±0.030 (0.341±0.053) |
| | M2REMAP Wen et al. (2023) | 0.873±0.004 (0.627±0.019) | 0.811±0.015 (0.549±0.026) | 0.689±0.028 (0.353±0.044) |
| | Π-model Rasmus et al. (2015) | 0.876±0.005 (**0.634**±0.014) | 0.819±0.020 (0.547±0.056) | 0.676±0.037 (0.347±0.054) |
| | Semi-VAE Zhang & Lu (2019) | 0.865±0.004 (0.623±0.018) | 0.809±0.011 (0.537±0.035) | 0.681±0.015 (0.339±0.047) |
| | **CaSBRE (Ours)** | **0.879**±0.003 (0.633±0.013) | **0.831**±0.009 (**0.573**±0.037) | **0.705**±0.022 (**0.370**±0.031) |
| 80% | S:DeepDDI Ryu et al. (2018) | 0.873±0.012 (0.558±0.027) | 0.890±0.031 (0.702±0.067) | 0.724±0.057 (0.419±0.093) |
| | S:MolTransHuang et al. (2021) | 0.883±0.009 (0.581±0.009) | 0.881±0.028 (0.690±0.051) | 0.725±0.049 (0.416±0.078) |
| | S:KGE_NFMYe et al. (2021) | 0.885±0.006 (0.599±0.012) | 0.843±0.124 (0.665±0.056) | 0.731±0.053 (0.412±0.069) |
| | M1 model Kingma et al. (2014) | 0.892±0.005 (**0.614**±0.013) | 0.901±0.021 (0.723±0.044) | 0.717±0.046 (0.391±0.099) |
| | M2REMAP Wen et al. (2023) | 0.889±0.009 (0.608±0.017) | 0.905±0.016 (0.720±0.050) | 0.739±0.052 (0.418±0.073) |
| | Π-model Rasmus et al. (2015) | 0.892±0.004 (0.611±0.017) | 0.902±0.024 (0.723±0.043) | 0.744±0.047 (0.429±0.073) |
| | Semi-VAE Zhang & Lu (2019) | 0.891±0.007 (0.611±0.011) | 0.907±0.013 (0.720±0.035) | 0.728±0.035 (0.405±0.078) |
| | **CaSBRE (Ours)** | **0.895**±0.002 (0.613±0.004) | **0.913**±0.020 (**0.725**±0.035) | **0.753**±0.041 (**0.472**±0.043) |

Table 2: **Results on the gene-disease association predictions** with AUROC (AUPRC) reported. S: supervised approach.

| Setting | Method | Random-pair | Single-cross | Double-cross |
|---|---|---|---|---|
| 20% | S:DeepDDI Ryu et al. (2018) | 0.825±0.004 (0.573±0.005) | 0.757±0.010 (0.524±0.016) | 0.592±0.011 (0.288±0.020) |
| | S:MolTrans Huang et al. (2021) | 0.821±0.008 (0.565±0.009) | 0.748±0.014 (0.517±0.025) | 0.592±0.019 (0.283±0.024) |
| | S:KGE_NFM Ye et al. (2021) | 0.821±0.008 (0.572±0.003) | 0.758±0.013 (0.525±0.022) | 0.628±0.018 (0.290±0.022) |
| | M1 model Kingma et al. (2014) | 0.846±0.003 (0.623±0.007) | 0.809±0.006 (0.537±0.008) | 0.614±0.012 (0.265±0.008) |
| | M2REMAP Wen et al. (2023) | 0.838±0.021 (0.609±0.011) | 0.805±0.018 (0.528±0.019) | 0.641±0.018 (0.288±0.021) |
| | Π-model Rasmus et al. (2015) | 0.847±0.002 (0.607±0.006) | 0.810±0.007 (**0.538**±0.011) | 0.634±0.014 (0.272±0.020) |
| | Semi-VAE Zhang & Lu (2019) | 0.839±0.010 (0.618±0.008) | 0.810±0.010 (0.529±0.021) | 0.626±0.010 (0.265±0.011) |
| | **CaSBRE (Ours)** | **0.850**±0.014 (**0.629**±0.007) | **0.812**±0.005 (0.537±0.012) | **0.645**±0.010 (**0.291**±0.013) |
| 80% | S:DeepDDI Ryu et al. (2018) | 0.815±0.005 (0.535±0.017) | 0.765±0.014 (0.517±0.038) | 0.645±0.037 (0.286±0.059) |
| | S:MolTrans Huang et al. (2021) | 0.820±0.008 (0.540±0.009) | 0.761±0.015 (0.506±0.033) | 0.652±0.022 (0.288±0.047) |
| | S:KGE_NFM Ye et al. (2021) | 0.822±0.004 (0.574±0.015) | 0.774±0.015 (0.521±0.036) | 0.650±0.064 (0.331±0.083) |
| | M1 model Kingma et al. (2014) | 0.859±0.002 (0.547±0.012) | 0.815±0.008 (0.523±0.014) | 0.661±0.047 (0.320±0.095) |
| | M2REMAP Wen et al. (2023) | 0.859±0.003 (0.548±0.004) | 0.809±0.012 (0.519±0.010) | 0.664±0.051 (0.323±0.045) |
| | Π-model Rasmus et al. (2015) | 0.866±0.004 (0.557±0.006) | 0.814±0.006 (0.525±0.011) | 0.650±0.041 (0.306±0.084) |
| | Semi-VAE Zhang & Lu (2019) | 0.857±0.004 (0.535±0.010) | 0.802±0.005 (0.518±0.007) | 0.668±0.033 (0.295±0.078) |
| | **CaSBRE (Ours)** | **0.872**±0.004 (**0.581**±0.014) | **0.822**±0.009 (**0.532**±0.014) | **0.681**±0.048 (**0.368**±0.091) |

in Equation 1 and $\mathcal{L}_{kl}$ in Equation 2; (iii) $S$-$C$ disentanglement objective $\mathcal{L}_{sc}$ in Equation 3. The overall encoder loss function is thus formulated as:

$$\mathcal{L}_{enc} = \mathcal{L}_{inter} + \sum_{*\in\{X,W\}} \alpha\mathcal{L}_{con}^* + \beta\mathcal{L}_{kl}^* + \gamma\mathcal{L}_{sc}^*, \tag{5}$$

where $\alpha$, $\beta$, and $\gamma$ are hyperparameters that balance the importance of the three objectives. Algorithm 1 outlines the interactive training procedure. To minimize the influence of spurious features, we perform *do-calculus* relation inference as described in Algorithm 2.

# 4 EXPERIMENTS

## 4.1 SETUP

### 4.1.1 DATASETS

We evaluate the methods on two representative interaction learning tasks, chemical-protein interactions (CPIs) and gene-disease associations (GDAs).

**Chemical-protein interactions** We evaluate CPIs based on annotations from Drugbank Wishart et al. (2018). There are 8,393 chemical-protein pairs, comprising 829 chemical molecules and 1,524 proteins. The molecules are represented using PubChem fingerprint, which indicates the presence of 881 substructures, and the proteins using pre-trained ESM2 protein embedding vectors Lin et al. (2023). We include 11,269 unlabeled chemical molecules and 20,504 unlabeled proteins during training.

**Gene-disease associations** We evaluate GDAs using annotations from DisGeNET Piñero et al. (2020). There are 38,938 labeled gene-disease pairs, encompassing 6,637 genes and 2,916 diseases.

The genes are represented by the protein embedding vectors derived from the pre-trained ESM2 Lin et al. (2023), and the diseases are represented using embedding vectors trained from large-scale electronic healthcare records Hong et al. (2021), which effectively capture the clinical semantic relationships between diseases. We include 20,504 unlabeled genes and 4,913 unlabeled diseases during training.

### 4.1.2 EVALUATIONS SETTINGS

We comprehensively evaluate the generalization of the interaction learning models in three settings. (i) **Random-pair**, in which we randomly split the annotations into train/validation/test pairs. (ii) **Single-cross**, in which we split by the entities from one domain and require that the train/validation/test data do not have overlapped entities in this domain. It evaluates the generalization to unseen entities of one domain. (iii) **Double-cross**, in which we split the entities from both domains into train/validation/test entities and ensure they do not have overlapping entities in either domain. We split the dataset into training, validation, and test sets in ratios of 2:4:4 (20% setting) and 8:1:1 (80% setting) for comprehensive evaluation of model generalization. To measure the performance, we report the area under the receiver operating characteristic curve (AUROC) and the area under the precision-recall curve (AUPRC). The results are averaged across five random splits of the training, validation, and test data. We further provide results for sensitivity analyses of hyperparameters and results on an extreme 5% setting in the Appendix.

### 4.1.3 IMPLEMENTATION DETAILS

The encoders and decoders from both domains are multi-layer perceptron networks with two hidden layers with 128 units, and 64-dimensional representations are learned for each domain. The $S$-$C$ predictors contain one hidden layer with 64 units. We set the threshold values $t_x$ and $t_w$ of the $S$-$C$ conditional discrepancy to be 0.5. We set the weight of the reconstruction loss $\alpha = 1.0$ and the prior distribution regularization $\beta = 0.5$. We progressively importance of $\mathcal{L}_{sc}$ and set $\gamma = 0.8 * (\frac{2}{1+\exp(-10 \cdot h)} - 1)$, where $h$ denotes the training progress ranging from 0 to 1. We provide more details on hyperparameter selection in the Supplementary Materials.

### 4.1.4 COMPARED METHODS

We make comparisons from the following three key perspectives: (i) We compare to the state-of-the-art (SOTA) supervised methods, DeepDDI Ryu et al. (2018) which models the interactions via a multi-layer perception network, MolTrans Huang et al. (2021) which captures the interactions through convolutional neural networks, and KGE_NFM Ye et al. (2021) which applies recommendation-system techniques to learn relations. (ii) We also compare our method to two general semi-supervised approaches: M1 model Kingma et al. (2014) which is pre-trained using VAE, and the II-model Rasmus et al. (2015) which ensures representation consistency across network permutations via Dropout. (iii) Two SOTA semi-supervised bio-medical extraction approaches are additionally compared: Semi-VAE Zhang & Lu (2019) which performs representation learning based on VAE, and M2REMAP Wen et al. (2023) which employs distribution matching to improve the generalization of relation inference to unlabeled entities. To ensure fair comparison, the input features for all comparison methods are the same as CaSBRE.

## 4.2 COMPARISON RESULTS

### 4.2.1 RESULTS OF CPIs

The results of the CPIs, presented in Table 1, demonstrate that CaSBRE consistently outperforms the compared methods in both AUROC and AUPRC metrics. As the proportion of labeled data increases from 20% to 80% for training, all methods show significant improvement in the *single-cross* and *double-cross* settings, though only comparable performance is seen in the *random-pair* setting. This suggests that these methods are prone to overfitting when the labeled data size is small. Compared to supervised approaches like DeepDDI, MolTrans, and KGE_NFM, the semi-supervised methods perform better in the *random-pair* setting but offer only similar performance in the *single-cross* and *double-cross* scenarios, indicating that previous semi-supervised methods do not significantly enhance generalization to unseen entities.

Table 3: **Ablation study** of the key components of CaSBRE.

| Method | Single-cross | Double-cross |
|---|---|---|
| CaSBRE (baseline) | 0.905±0.020 (**0.735**±0.051) | 0.724±0.019 (0.422±0.056) |
| CaSBRE (w/o $\mathcal{L}_{sc}$) | 0.905±0.027 (0.733±0.047) | 0.725±0.064 (0.419±0.109) |
| CaSBRE (w/o *do-calculus*) | 0.910±0.014 (0.720±0.030) | 0.730±0.058 (0.441±0.097) |
| CaSBRE | **0.913**±0.020 (0.725±0.035) | **0.753**±0.041 (**0.472**±0.043) |

The advantages of CaSBRE are substantial both in the *single-cross* and *double-cross* settings. In the setting with 20% labels, CaSBRE outperforms the second-best method by 1.5% in AUROC and 4.8% in AUPRC for the *single-cross* and 3.4% and 2.8%, respectively, for the challenging *double-cross*. With an increased label size to 80% of the total annotations, the advantages of CaSBRE still persist, attaining an improvement of 1.2% in AUROC and 10.0% in AUPRC over the second-best models in *double-cross* scenario.

### 4.2.2 Results of GDAs

The results of GDAs are shown in Table 2. For the three settings, increasing the labels from 20% to 80%, only minor improvements are consistently observed across all the methods. This is attributed to the scale of combinations between the 6,637 genes and 2,916 diseases being significantly larger than those in CPIs and the total annotations. The semi-supervised methods-M1 model, II-model, Semi-VAE, and M2REMAP-consistently bring improvements in the three settings over the supervised approaches by leveraging the extra unlabeled data.

CaSBRE shows substantial performance advantages over the semi-supervised methods such as II-model and Semi-VAE in both the *random-pair* and *single-cross* and the advantages become consistently distinct in the *double-cross*. For example, with 80% labels, CaSBRE outperforms the II-model by 0.7% in AUROC and 3.3% in AUPRC. Further, the advantages of CaSBRE over the SOTA-supervised methods grow substantially; for example, with 80% labels, outperforming the DeepDDI by 5.0% in AUROC and by 13.4% in AUPRC on average across the three settings.

### 4.3 Ablation Studies

In Table 3, we provide ablation studies on the chemical-protein relation prediction task using 80% of the labels for training. The combination of $\mathcal{L}_{sc}$ and *do-calculus* brings distinct improvements over the baseline model, by 4.0% in AUROC and 11.8% in AUPRC, in the *double-cross* scenario while in the *single-cross* the benefits are minor. Notably, with $\mathcal{L}_{sc}$ only or *do-calculus* only performance gains are ensured, which is reasonable, because *do-calculus* is built upon well-disentangled spurious features from the causal ones.

### 4.3.1 Representation Visualization

Figure 2 presents a visualization of the $S$-$C$ predictions using principal component analysis to illustrate feature disentanglement on the CPIs relation inference. To compare the two domains, we conduct experiments in the *single-cross* (proteins) setting and visualize the $C$ values from both proteins and chemicals. As shown in Figure 2, for proteins, the $C$ values generated by the feature encoder closely align with the predicted $C$ values from the $S$-$C$ predictor in the labeled training data, while they are clearly separated in the unlabeled test data. This indicates that the $S$-$C$ predictor effectively detects discrepancies in $S$-$C$ dependency between labeled and unlabeled protein data. Meanwhile, for chemicals, the $C$ values are well-aligned in both labeled training and unlabeled test data, which is expected because the training and test data are under the same distribution for chemicals in this *single-cross* setting.

### 4.3.2 $S$-$C$ Loss Visualization

Figure 3 illustrates the $S$-$C$ disentanglement loss ($\mathcal{L}_{sc}$), the interaction loss ($\mathcal{L}_{inter}$), and the Area Under the Receiver Operating Characteristic (AUROC) on the test data. Around the 200*th* training step, $\mathcal{L}_{sc}$ experiences a significant decline, which correlates with a decrease in performance and an increase in $\mathcal{L}_{inter}$. This is because $\mathcal{L}_{sc}$ is designed to guide the encoders in reorganizing the learned

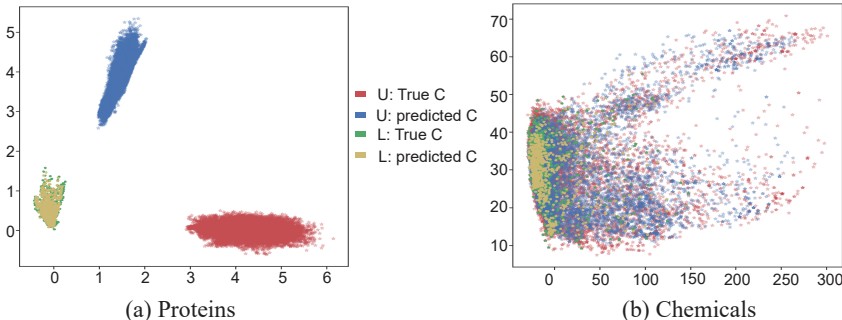

(a) Proteins                     (b) Chemicals

Figure 2: **Visualization of $S$-$C$ predictions** of the labeled training (L) and unlabeled test (U) entities on the CPIs task. The *true* $C$ denotes the $C$ obtained by the encoders and the *predicted* by the $S$-$C$ predictor **P**.

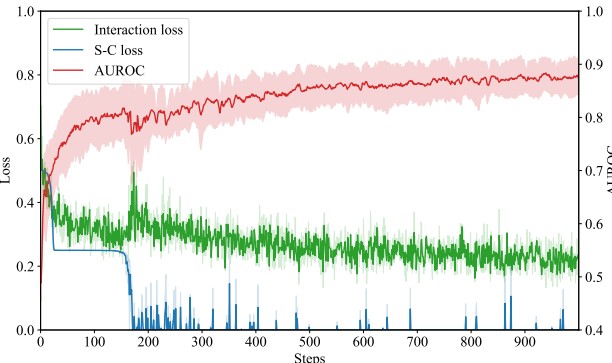

Figure 3: Visualization of the interaction loss $\mathcal{L}_{inter}$, S-C loss $\mathcal{L}_{sc}$, and AUROC on test data.

representation into causal and spurious components, temporarily hindering interaction learning. Subsequently, the $\mathcal{L}_{sc}$ stabilizes, becoming smaller on average and exhibiting fewer fluctuations, which indicates improved feature disentanglement. It is noteworthy that, since the $S$-$C$ predictor is trained interactively with the encoders, the fluctuations in $\mathcal{L}_{sc}$ are to be expected.

## 5 CONCLUSION

In this paper, we introduce CaSBRE, a causality-inspired semi-supervised framework designed to enhance the accuracy and generalizability of biomedical relation extraction models. By effectively disentangling causal from spurious features, CaSBRE mitigates the impact of misleading correlations, resulting in robust predictions across two representative biomedical datasets. The framework's superiority in handling scarce labeled data and its capacity to generalize to unseen entities are particularly crucial for advancing biomedical research and therapeutic development.

Looking ahead, we aim to extend CaSBRE's methodologies to other machine learning paradigms, such as transfer learning and active learning. These explorations will help refine the model's efficacy and adaptability, ensuring that it not only addresses the challenges in biomedical relation extraction but also contributes broadly to the field of semi-supervised learning. Additionally, future work can leverage the insights gained from our findings to enhance the ability of machine learning models to make accurate predictions in dynamically changing environments with limited labeled data.

Our work has several limitations. First, while CaSBRE is a causality-inspired method, causal conclusions should be made with caution, especially in the biomedical domain. Furthermore, during $S$-$C$ disentanglement, CaSBRE implicitly incorporates the interaction effects between the two domains by joint optimization with $\mathcal{L}_{inter}$. Future work may consider more sophisticated feature disentanglement explicitly involving two domains, potentially using more complex causal graphs.

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

## A   CASBRE ALGORITHM

---

**Algorithm 1 Training of CaSBRE**

---

**Require:**
1: $D_x, D_w$: Embeddings of labeled entities $x$ and $w$
2: $Y_{xw}$: Binary relation labels between $x$ and $w$
3: $D_{x_u}, D_{w_u}$: Embeddings of unlabeled entities $x_u$ and $w_u$
4: $\alpha, \beta, \gamma$: Hyperparameters for loss weighting
5: $t$: Threshold for S-C disentanglement loss
**Ensure:**  Optimal parameters $\theta = \{\mathbf{En}_x, \mathbf{En}_w, \mathbf{De}_x, \mathbf{De}_w, \mathbf{M}\}$ and $\phi = \{\mathbf{P}_x, \mathbf{P}_w\}$
6: Initialize network parameters $\theta$ and $\phi$
7: **repeat**
8:     Sample minibatch $(x, w, y_{xw})$ from $(D_x, D_w, Y_{xw})$
9:     Sample minibatch $(x_u, w_u)$ from $(D_{x_u}, D_{w_u})$
10:    // Encode causal (c) and spurious (s) representations for all entity types
11:    **for** $e \in \{x, w, x_u, w_u\}$ **do**
12:        $(\mu_e^c, \sigma_e^c), (\mu_e^s, \sigma_e^s) \leftarrow \mathbf{En}_e(e)$
13:        $c_e \leftarrow$ sample from $\mathcal{N}(\mu_e^c, \sigma_e^c)$
14:        $s_e \leftarrow$ sample from $\mathcal{N}(\mu_e^s, \sigma_e^s)$
15:    **end for**
16:    // Compute reconstruction loss
17:    $\mathcal{L}_{con} \leftarrow \sum_{e \in \{x, w, x_u, w_u\}} \|e - \mathbf{De}_e(c_e, s_e)\|_2$
18:    // Compute KL divergence loss (for variational inference)
19:    $\mathcal{L}_{kl} \leftarrow -\frac{1}{2} \sum_{e \in \{x, w, x_u, w_u\}} \sum_{v \in \{c, s\}} (1 + 2 \log \sigma_e^v - (\mu_e^v)^2 - (\sigma_e^v)^2)$
20:    // Compute interaction loss
21:    $\hat{y}_{xw} \leftarrow \sigma((c_x, s_x) \cdot \mathbf{M} \cdot (c_w, s_w)^\top)$ {Sigmoid applied to the prediction score}
22:    $\mathcal{L}_{inter} \leftarrow -y_{xw} \log(\hat{y}_{xw}) - (1 - y_{xw}) \log(1 - \hat{y}_{xw})$
23:    // Compute S-C prediction loss
24:    $\mathcal{L}_c \leftarrow \sum_{e \in \{x, w\}} \|\mathbf{P}_e(s_e) - c_e\|_2$
25:    // Compute S-C disentanglement loss
26:    $\mathcal{L}_{sc} \leftarrow \sum_{e \in \{x, w\}} \max(\|\mathbf{P}_e(s_e) - c_e\|_2 - \|\mathbf{P}_e(s_{e_u}) - c_{e_u}\|_2 + t, 0)$
27:    // Update parameters using gradient descent
28:    $\mathcal{L}_{enc} \leftarrow \mathcal{L}_{inter} + \alpha \mathcal{L}_{con} + \beta \mathcal{L}_{kl} + \gamma \mathcal{L}_{sc}$
29:    $\theta \leftarrow \theta - \eta_\theta \nabla_\theta \mathcal{L}_{enc}$ {Update encoder-decoder and interaction learning parameters}
30:    $\phi \leftarrow \phi - \eta_\phi \nabla_\phi \mathcal{L}_c$ {Update S-C predictor parameters}
31: **until** convergence criteria met

---

---

**Algorithm 2 Relation inference of CaSBRE.**

---

**Require:** Query embedding $x$ and $w$
1: $(\mu_e^c, \sigma_e^c), (\mu_e^s, \sigma_e^s) \leftarrow \mathbf{En}_e(e), e \in \{x, w\}$;
2: $c_e \leftarrow \mathcal{N}(\mu_e^c, \sigma_e^c), e \in \{x, w\}$;
3: **for** $i = 1, ..., K$ **do**
4:    $s_x^i, s_w^i \leftarrow \mathcal{N}(0, 1)$;
5:    $\hat{y}^i \leftarrow \text{Sigmoid}\left((c_x, s_x^i)\mathbf{M}(c_w, s_w^i)^T\right)$ ;
6: **end for**
7: $\hat{y} \leftarrow \frac{1}{K} \sum_{i=1}^{K} \hat{y}^i$ ;

---

## B   ADDITIONAL RESULTS

### B.1   RESULTS ON EXTREMELY SMALL TRAINING SET

Table 4 shows the results on extremely small labeled data (5% setting), alongside comparisons with the fully supervised model DeepDDI and the semi-supervised Π-model. The results further highlight CaSBRE's consistent superiority in generalizing to unseen entities, particularly under limited supervision.

Table 4: Performance Comparison with Extremely Limited Labeled Data (5%)

| Dataset | Method | Random-pair | Single-cross | Double-cross |
|---------|--------|-------------|--------------|--------------|
| CPI | DeepDDI | 0.752 (0.458) | 0.705 (0.396) | 0.588 (0.236) |
| | Π-model | 0.769 **(0.474)** | 0.711 (0.418) | 0.604 (0.255) |
| | CaSBRE | **0.771** (0.472) | **0.729 (0.423)** | **0.623 (0.266)** |
| GDA | DeepDDI | 0.770 (0.488) | 0.722 (0.416) | 0.558 (0.221) |
| | Π-model | 0.770 (0.493) | 0.729 (0.439) | 0.563 (0.230) |
| | CaSBRE | **0.772 (0.496)** | **0.738 (0.454)** | **0.580 (0.247)** |

## B.2 SENSITIVITY ANALYSIS

To evaluate the robustness of CaSBRE, we conducted comprehensive sensitivity analyses on four key hyperparameters: $\alpha$, $\beta$, $\gamma$ in equation 5, and the threshold $t$ (where we use $t_x = t_w = t$) in the S-C loss, as well as the latent space dimension, on the CPI prediction task (20% labels, double-cross setting). The results demonstrate that CaSBRE exhibits robust performance across different hyperparameter settings (Table 5,6,7,8). We further investigated the effect of latent space dimensionality on model performance, as shown in Table 9. It shows that a moderate latent dimension yields the best performance. Extremely low or high dimensions tend to underfit or overfit the labels.

Table 5: Sensitivity analysis of parameter $\alpha$

|        | $\alpha = 0.5$ | $\alpha = 1.0$ | $\alpha = 1.5$ |
|--------|-------|-------|-------|
| auROC  | 0.701 | 0.705 | 0.702 |
| auPRC  | 0.369 | 0.370 | 0.365 |

Table 6: Sensitivity analysis of parameter $\beta$

|        | $\beta = 0.25$ | $\beta = 0.5$ | $\beta = 0.75$ |
|--------|-------|-------|-------|
| auROC  | 0.705 | 0.705 | 0.697 |
| auPRC  | 0.364 | 0.370 | 0.352 |

## C LLM USAGE DISCLOSURE

We utilized a large language model (LLM) to assist with proofreading during the preparation of this manuscript. The LLM was used to improve grammar, clarity, and readability. All final content, including all scientific claims and conclusions, was authored and verified by us, and we take full responsibility for the paper's content.

Table 7: Sensitivity analysis of parameter $\gamma$

|  | $\gamma =$ dynamic | $\gamma = 0.6$ | $\gamma = 0.8$ | $\gamma = 1.0$ |
|---|---|---|---|---|
| auROC | 0.705 | 0.701 | 0.703 | 0.702 |
| auPRC | 0.370 | 0.362 | 0.361 | 0.367 |

Table 8: Sensitivity analysis of threshold parameter $t$

|  | $t = 0.25$ | $t = 0.5$ | $t = 0.75$ |
|---|---|---|---|
| auROC | 0.701 | 0.705 | 0.704 |
| auPRC | 0.362 | 0.370 | 0.370 |

Table 9: Effect of latent space dimension on CaSBRE performance

| Latent Dimension | 8 | 16 | 32 | 64 | 128 |
|---|---|---|---|---|---|
| auROC | 0.687 | 0.701 | 0.702 | 0.705 | 0.688 |
| auPRC | 0.345 | 0.370 | 0.372 | 0.370 | 0.364 |

