# OpenReview forum: "CaSBRE: Causality-inspired Semi-supervised Biomedical Relation Extraction"
_ICLR.cc/2026/Conference — Submitted to ICLR 2026_

### Official Review · Reviewer_Ukja · 2025-10-25

**Soundness:** 2
**Presentation:** 3
**Contribution:** 2
**Rating:** 4
**Confidence:** 3

**Summary:**

The paper CaSBRE (Causality-Inspired Semi-Supervised Biomedical Relation Extraction) proposes a causality-driven framework to improve biomedical relation extraction, such as chemical–protein and gene–disease interactions, under limited labeled data. Traditional semi-supervised methods often overfit to dataset-specific biases, so CaSBRE introduces a Feature Disentanglement Network to separate causal from spurious features by detecting inconsistencies between labeled and unlabeled data, and a do-calculus inference module to marginalize out spurious effects during prediction. Tested on DrugBank (CPI) and DisGeNET (GDA) datasets, CaSBRE significantly outperforms existing supervised and semi-supervised baselines, especially when generalizing to unseen biomedical entities. This demonstrates that causality-inspired disentanglement and inference can substantially enhance robustness and generalization in biomedical relation learning.

**Strengths:**

1. The paper accurately identifies two core challenges in current biomedical relation extraction, which are *training set distribution bias (domain bias)* and *label scarcity*.
2. The proposed framework is intuitive and easy to reproduce, without relying on large external modules or complex graph structures.

3. The writing follows a coherent logical structure, and the conclusion section provides a thoughtful discussion of the method’s limitations.

**Weaknesses:**

1. **Weak causal foundations**

   The framework lacks an explicit causal graph (DAG), structural causal model (SCM), or counterfactual formulation. The “do-calculus” step is only a heuristic averaging procedure rather than a theoretically identifiable intervention. The key assumption $P_l(C|S) \neq P_u(C|S)$ is also not empirically verified. The approach is more like a statistical learning framework wrapped by causality.

2. **Lack of interpretability in feature disentanglement**

   The paper claims to disentangle causal (C) and spurious (S) features, but provides no semantic or biological interpretation of these components. Figure 2 only shows point-cloud separation without explaining what the dimensions mean or whether (C) aligns with known biochemical properties.

3. **Heuristic implementation of do-calculus**

   The “intervention” simply samples $S \sim N(0,1)$ and averages predictions. This prior is a little arbitrary. The paper does not discuss how the sampling number $K$ or prior choice affects stability or performance. This step behaves more like stochastic ensembling than formal causal inference.

4. **Limited experiments**

   Experiments are restricted to two datasets (CPI and GDA), and the baselines are earlier approaches, rather than the most recent. The ablation only removes $L_{sc}$ or do-calculus, without comparing against alternative disentanglement or SSL strategies. The generality of the conclusions is therefore limited.

5. **Lack of biological interpretability**

   While performance gains are shown, the paper does not analyze whether the learned causal features correspond to known biological mechanisms. No case studies or example-level explanations are provided, which weakens credibility for domain adoption.

**Questions:**

1. **Empirical support for the $P_l(C|S) \neq P_u(C|S)$ assumption**

   Can the authors provide quantitative or visual evidence showing that the conditional distributions indeed differ?

2. **Rationale for the do-calculus sampling prior**

   Why is $S \sim N(0,1)$ chosen? Have the authors attempted to estimate $P(S)$ from the data or tested the sensitivity to the prior and the number of samples $K$?

3. **Interpretation of the disentangled features**

   What do $C$ and $S$ correspond to biologically? Can the authors visualize attention maps, motifs, or clusters that support the claim of causal versus spurious components?

4. **Comparison with alternative semi-supervised methods**

   How does CaSBRE compare with modern SSL approaches such as consistency regularization, pseudo-labeling, or contrastive learning? Is the improvement due to the “causal” idea or simply an effect of domain regularization?

5. **Terminology accuracy**

   Given the lack of formal causal identifiability, would it be more precise to describe the method as a “distributionally robust semi-supervised framework” rather than “causality-inspired”?

6. **Generalization and extensibility**

   Can the framework be extended to multi-relational (n-ary) or multimodal biomedical knowledge graphs? The paper briefly mentions transfer and active learning as future work; elaborating a concrete path would strengthen the contribution.

---

> ### Author Response · Authors · 2025-11-27
>
> We thank Reviewer Ukja for the detailed review and for acknowledging the **importance of the problem** (bias and scarcity) and the **intuitive nature** of our framework. We appreciate the challenge regarding the rigorousness of our causal foundations, as it allows us to clarify the theoretical positioning of our work. Below, we address the concerns regarding causal theory, interpretability, and implementation choices.
> # Weak Causal Foundations (W1, Q5)
> We agree with the reviewer that we are not performing classical causal inference. This is why we use the term “causality-inspired” rather than “causal” in the paper’s title. The underlying SCM is that the Input $X$ consists of Causal factors $C$ and Spurious factors $S$ (confounders). $S$ influences the Label $Y$ only in the biased training distribution (due to selection bias), creating a backdoor path $X \leftarrow S \rightarrow Y$. However, the task of **interaction learning** makes it more **challenging** because it learns the C and S under a **pairwise interaction scenario** ($C_x, C_w, S_x, S_w$). This leads to the uniqueness and novelty of our framework.
>
> We use the term “causality-inspired” because we operationalize the **Backdoor Criterion**. By disentangling $S$ and marginalizing it out, we simulate an intervention $do(X)$, cutting the dependence on the confounder.
>
> While "Distributionally Robust" is a fair descriptor, we believe "Causality-inspired" is precise because the mechanism for achieving robustness is explicitly derived from the Pearlian adjustment formula, even if the variables are learned latent vectors rather than observed scalars.
>
> # Empirical Support for the Assumption (Q1)
>
> Figure 2 provides an empirical verification of the assumption that S-C correlations in the labeled data may not generalize to unlabeled data. Under this single-cross setting (cross protein), we have shown that S-C predictor successfully predicts C from S for all labeled data, but fails to predict for unlabeled protein. This visual divergence confirms that the conditional distributions indeed differ, validating our strategy of maximizing this discrepancy to identify spurious features.
>
> # Rationale for the do-calculus sampling (W3, Q2)
> The choice of Gaussian prior is not arbitrary; it is commonly used in the VAE architecture. For stability, we found that the method is stable with $K=20$ samples. Increasing $K$ beyond this yielded diminished returns.
>
> # Interpretability (W2, W5, Q3)
> While interpreting specific dimensions of dense embeddings is challenging, we offer the following hypothesis regarding the nature of S and C in this domain. S can be the popularity or research bias in the training set, and C can be the actual physicochemical compatibility required for a true interaction. By forcing the model to ignore S via do-calculus, we improve performance on unseen entities, as shown in the ablation study comparing CaSBRE with or without do-calculus in Table 3. This suggests that CaSBRE is now relying on the actual biochemical features (C).
>
> # Comparison to modern SSL (Q4)
> We compared against M2REMAP, which is a very recent SSL method specifically for biomedical relation extraction. Generic SSL methods often rely on data augmentation (e.g., cropping/rotating images). As noted in our Introduction, perturbing molecular graphs or protein sequences (augmentations) often destroys their biological validity, making these methods difficult to apply directly without domain-specific modification. CaSBRE is designed specifically for this paired, non-augmentable domain.
>
> # Generalization (Q6)
> Yes. The core principle—using the discrepancy between labeled and unlabeled correlations to identify confounders—is agnostic to the data type. For a KG, the VAE encoders may be replaced by GNNs, and the S-C predictor would operate on node embeddings. We will add a discussion on this extensibility to the Conclusion.
>
> We hope these clarifications address your concerns and warrant a reconsideration of the score.

---

### Official Review · Reviewer_LSEX · 2025-10-31

**Soundness:** 4
**Presentation:** 3
**Contribution:** 3
**Rating:** 6
**Confidence:** 4

**Summary:**

This paper introduces CaSBRE, a causality-inspired semi-supervised framework for biomedical relation extraction. It aims to disentangle and mitigate the impact of such spurious correlations by separating causal features from spurious ones. The core idea is to exploit the discrepancy in S-C correlations between labeled and unlabeled data as a supervisory signal for disentanglement. For inference, it employs a do-calculus interaction inference strategy to marginalize the influence of spurious features, thereby improving generalization to unseen entities.

**Strengths:**

- The paper proposes a novel feature disentanglement strategy that leverages the assumption that S-C correlations in labeled data may not generalize to unlabeled data.
- The empirical results are strong and consistent, showing that CaSBRE substantially outperforms state-of-the-art methods in the most challenging OOD settings.

**Weaknesses:**

- The paper lacks related work of causality-inspired representation learning for OOD generalization.
  - Arjovsky et al. "Invariant risk minimization." arXiv preprint arXiv:1907.02893 (2019).
  - Lv et al. "Causality inspired representation learning for domain generalization." Proceedings of CVPR. 2022.
- The ablation study in Table 3 is unclear; the configuration of CaSBRE (baseline) is not specified.
- The paper lacks a qualitative analysis of the learned spurious features (S), leaving it unclear what semantic information or dataset biases are actually being captured.
- The ablation study shows that feature disentanglement alone improves performance, but the paper does not provide a rationale for why this improvement occurs over the baseline.
- The paper does not justify the choice of marginalizing S via do-calculus over simpler alternatives, such as discarding S entirely and using only C for prediction.

**Questions:**

- Could you clarify the precise configurations for "CaSBRE (baseline)" ?
- How does your SSL-based approach for achieving invariance compare to other OOD generalization methods?
- Have you performed any qualitative analysis to interpret what semantic information or dataset biases the learned spurious features (S) are capturing?
- The ablation study shows that feature disentanglement alone (w/o do-calculus) improves performance over the baseline. What is the authors' hypothesis for this performance gain?
- Could you justify the choice of marginalizing S via do-calculus for inference, as opposed to a simpler alternative such as discarding S and using only C for prediction?
- Could you comment on the stability and convergence of the training procedure (Algorithm 1), particularly its sensitivity to initialization or hyperparameter choices?

---

> ### Author Response · Authors · 2025-11-27
>
> We sincerely thank Reviewer LSEX for the encouraging assessment and for recognizing the soundness of our approach (Rating: 4/4) and the strength of our empirical results. We appreciate the insightful questions regarding the rationale behind our specific causal mechanisms. Below, we provide clarifications on the baseline, the justification for our inference strategy, and the connections to OOD literature.
> # Clarifications on baselines and ablations (Q1, Q4, Q5)
> ## Q1: CaSBRE (baseline) configurations
> The "CaSBRE (baseline)" in Table 3 refers to the Variational Autoencoder (VAE) backbone combined with the Duplex Interaction Module, but without the proposed causal components, excluding both S-C disentangle loss and do-calculus. We have added detailed explanations in the revised version.
> ## Q4: Feature disentanglement alone improves the performance
> We hypothesis that the reason why feature disentanglement ($\mathcal{L}_{sc}$) improves performance even without do-calculus is that by explicitly penalizing the encoder when it learns correlations that exist in labeled data but not in unlabeled data, the encoder is forced to learn a "C" (Causal) representation that is more robust and invariant across distributions. This probably leads to a higher quality of C and S, which leads to better generalization even if the inference method is standard.
> ## Q5: Justify the choice of marginalizing S vs. discarding S
> We chose marginalization (averaging over the prior $P(S)$) rather than discarding $S$ for two reasons:
> + 1) The Feed-Forward Layers (FFL) in the interaction module are trained on the concatenated representation $[C, S]$. Completely removing $S$ (or zeroing it out) during inference would create a dimension mismatch or a drastic distribution shift in the input layer of the FFL, degrading performance.
> + 2) In our structural causal model, $S$ represents confounders (e.g., bias/context). The correct operation to simulate an intervention $do(S)$ is to cut the dependence of $S$ on the input instance, forcing $S$ to follow its prior distribution (random noise), while keeping the computational graph intact. This effectively "averages out" the effect of the spurious features rather than assuming they have zero weight.
> # Related work and OOD generalization (W1, Q2)
> ## W1 & Q2: Comparison to Invariant Risk MInimization (IRM) and CVPR paper
> We thank the reviewer for pointing out these related work. We will discuss them in the revised Related Work section. We argue that CaSBRE is distinct because it’s a causal inspired network for **paired** biomedical relation data.
>
> # Qualitative analysis and stability (Q3, Q6)
> ## Q3: Qualitative analysis of learned spurious features
> While interpreting specific dimensions of dense embeddings is challenging, we offer the following hypothesis regarding the nature of S and C in this domain. S can be the popularity or research bias in the training set, and C can be the actual physicochemical compatibility required for a true interaction. By forcing the model to ignore S via do-calculus, we improve performance on unseen entities, as shown in the ablation study comparing CaSBRE with or without do-calculus in Table 3. This suggests that CaSBRE is now relying on the actual biochemical features (C).
>
> Figure 2 further provides an empirical verification of the assumption that S-C correlations in the labeled data may not generalize to unlabeled data. Under this single-cross setting (cross protein), we have shown that S-C predictor successfully predicts C from S for all labeled data, but fails to predict for unlabeled protein. This visual divergence confirms that the conditional distributions indeed differ, validating our strategy of maximizing this discrepancy to identify spurious features.
>
> ## Q6: Stability and convergence of Algorithm 1
> Our training is stable. As shown in Figure 3, the interaction loss and S-C loss converge effectively after approximately 800 steps. We also conducted sensitivity analyses on hyperparameters (Tables 5-9 in the Appendix). It shows the robustness of CaSBRE to hyperparameter choices.

---

### Official Review · Reviewer_cdEd · 2025-10-31

**Soundness:** 2
**Presentation:** 2
**Contribution:** 2
**Rating:** 2
**Confidence:** 4

**Summary:**

This paper proposes a novel pipeline for relation extraction between entities (e.g., gene-disease, chemical interactions) with the goal of moving beyond spurious correlations to identify more causal relationships. The core contribution is a two-step approach: first, a Variational Autoencoder (VAE) is used to learn representations that aim to disentangle causal factors from spurious ones. Second, during inference, a marginalization step inspired by the backdoor criterion is applied to control for confounding bias. The authors claim this method improves the robustness and accuracy of relation extraction by focusing on underlying causal mechanisms. To validate their approach, they test the pipeline on several comprehensive datasets, presenting competitive results for most cases.

**Strengths:**

- Comprehensive Evaluations: The paper is supported by comprehensive evaluations, including ablation studies on the model's components and helpful visualizations, which add credibility to the empirical results.

- Causality-Inspired Approach: Tackling problems with huge hypothesis spaces and limited labeled data, such as relation extraction, is a promising direction, and framing the problem from a causal perspective is a usually valuable .

**Weaknesses:**

The paper's weaknesses can be categorized into three main areas: text and notations, novelty, and engagement with related work.

## Related Work

This is the main drawback of the paper.
* **Lack of Causal Inference Context:** For a paper centered on a causality-inspired method, the related work section is surprisingly sparse on the topic. It fails to situate the work within the vast and long-explored field of distinguishing spurious correlations from causal relations. There is no introduction to the specific definitions of causality being used or a comparison to alternative methods.
* **Misleading Focus and Claims:** The related work focuses on SSL methods for causal discovery but without sufficient citations. The claim that "Most current methods rely on supervised methods" is inaccurate (see for example the large body of literature on topics like GRN and PPI inference uses unsupervised methods on batch data.)
* **Superficial Treatment of Causal Concepts:** Key concepts like the backdoor criterion are mentioned casually, without stating the actual criteria or acknowledging the significant challenges and assumptions (e.g., causal sufficiency) required for its valid application, which are often not met in real-world data. The vast literature on the limitations of causal learning is not addressed at all.

**(c) Novelty**

* **Limited Methodological Novelty:** While the goal is commendable, the novelty of the approach is limited. The pipeline combines existing components (VAE, backdoor-inspired marginalization), but there is a major lack of a deep, rigorous dive into these methods and their known limitations.
* **Unsupported Claims of Novelty:** The paper claims the "introduction of a benchmark dataset" as a contribution, yet the datasets mentioned (chemical-protein, gene-disease) are widely used benchmarks in the community. The authors need to clarify what makes their specific formulation a novel contribution.
* **Overstated Framework Novelty:** The claim of proposing the "first general semi-supervised framework for causal interaction learning" is a significant overstatement. Causality-inspired methods are prevalent in related biomedical fields, such as GRN inference, with many existing frameworks that could be considered precedents.

## Text and Notations

* **Clarity and Structure:** The text is difficult to follow. While the motivation and preliminaries are relatively well-written, the core details of the proposed framework are repeatedly mentioned in a high-level manner up to page 4, making it hard to grasp the specifics. The main figure, which should provide a clear overview, is not introduced until page 4 and is accompanied by minimal explanation.
* **Repetition:** Repetition is a recurring issue throughout the paper (see for example the Introduction and Section 2.2).
* **Unclear Notations:** Several key notations remain ambiguous. For instance, in lines 091-095, the origin and meaning of `C` are confusing, being referred to as derived from both labeled and unlabeled data. The exact meaning of `P(C|S)` is not defined: Is `C` a subset of `S`? Are `C` and `S` distinct sets of correlations? Is the analysis performed on univariate relations or in a multivariate context? These ambiguities hinder a full understanding of the method.

**Questions:**

See Weaknesses

---

> ### Author Response · Authors · 2025-11-27
>
> We thank Reviewer cdEd for the detailed and constructive feedback. We value the acknowledgement of our **comprehensive evaluations** and the agreement that **framing relation extraction from a causal perspective is a valuable direction**. We have taken the criticism regarding Related Work, Novelty, and Clarity very seriously. Below, we address each point and outline the substantial revisions we have prepared to improve the paper.
>
> # Related work and causal inference context
> ## Q1: Lack of causal inference context
>
> We acknowledge this oversight. In the revision, we have significantly expanded Section 2 to include a dedicated subsection on "Causal Discovery and Spurious Correlations.” We explicitly define our causal perspective: we view the "Entity Pair $\to$ Relation" problem through a structural causal model where unobserved confounders (spurious features $S$) create a backdoor path between input features and labels. Our method acts as a proxy intervention.
>
> ## Q2: Misleading focus
> We appreciate this correction regarding the broader field of network inference. Our claim referred specifically to the sub-field of feature-based biomedical relation extraction, where supervised deep learning is indeed dominant (DeepDDI, MolTrans, etc.). We have rewritten this sentence to acknowledge unsupervised methods in GRN and PPIs. We clarify that our contribution is distinct: we introduce a **Semi-Supervised** framework that leverages both labeled and unlabeled data for feature-rich entity pairs, bridging the gap between purely supervised extraction and unsupervised network inference.
>
> ## Q3: Superficial Treatment of Causal Concepts
> We acknowledge that strictly ensuring causal sufficiency (no unobserved confounders) is impossible in this domain. We treat the learned latent space as a proxy for the total feature space. Our core assumption is that by disentangling the latent space into $C$ (causal) and $S$ (spurious) based on distribution shifts between labeled and unlabeled data, we effectively learn the confounders within the latent space. CaSBRE is a causality-**inspired** semi-supervised method. We have added more sentences in limitations to explicitly state that our do-calculus is an approximation contingent on the quality of the disentanglement.
>
> # Novelty
> ## Q4: Limited Methodological Novelty
> While VAEs and the backdoor criterion are existing tools, our novelty lies in how they are integrated to solve a specific biological problem.
> + First, the $\mathcal{L}_{SC}$ is a core novelty that identifies spurious features by detecting distribution shifts between labeled and unlabeled data. This is novel in both in general machine learning and in biomedical relation extraction.
> + Furthermore, unlike standard causal NLP tasks, we apply this to a paired input domain (Chemicals AND Proteins), requiring a dual-stream disentanglement which is structurally distinct from standard single-stream causal learning and is therefore **nontrivial**.
> ## Q5: Unsupported Claims of Novelty
> We apologize for the phrasing confusion. Beyond existing relational data, we have collected a set of unlabeled entities with their embedding for benchmarking semi-supervised algorithms. Moreover, we established a benchmark evaluation setting (specifically the **Single-cross** and **Double-cross** splits) on these datasets. This rigorous splitting protocol that tests out-of-distribution generalization. We have rephrased this contribution to explicitly state the differences from existing datasets.
>
> ## Q6: Overstated Framework Novelty
> Thank you for mentioning this. To the best of our knowledge, this is the first causality-inspired interaction learning model for biomedical relation extraction using entity embeddings. We have revised the claim to make it more specific.
>
> # Text and Notations
> ## Q7: Clarity and Unclear Notations
> As described in Section 3.3.1, $C$ and $S$ are not subsets of correlations. They are distinct sub-vectors of the latent representation $Z$. **$P(C|S)$** represents the statistical dependency between the causal and spurious dimensions within the latent space. In the labeled data (biased), $S$ allows us to predict $C$ (high correlation). In the unlabeled data (unbiased/general), this correlation breaks down. Our loss function minimizes this dependency in the "general" distribution while acknowledging it in the "biased" distribution.

---

### Official Review · Reviewer_hsBr · 2025-11-01

**Soundness:** 3
**Presentation:** 3
**Contribution:** 3
**Rating:** 6
**Confidence:** 4

**Summary:**

Prediction of biomedical interaction relations such as chemical-protein interactions and gene-disease associations is important for biomedical research. However, due to the scarcity and bias of annotated interaction relation, it is hard for the model to capture the generalizable features for interaction relation prediction. This paper proposes a semi-supervised learning framework to disentangle the spurious correlations from real interactions and perform additional do-calculus inference to further reduce the impact of spurious correlations on the interaction relation prediction. The experimental results show that the proposed CaSBRE method outperforms other supervised and semi-supervised models in many settings, especially in setting with unseen biomedical entities.

**Strengths:**

(1) The paper is well written and well organized. It provides clear definition of the problem, clear explanation of motivation, and enough background knowledge to make the paper easy to follow.

(2) The CaSBRE method proposed in the paper is formalized rigorously and technically sound. The feature disentanglement network and causal interaction inference are described with both mathematical formalizations and corresponding algorithms.

(3) The paper conducts ablation study and addition experiments to verify the learning result of the proposed model is consistent with the design objective, which strengthens the paper quality.

**Weaknesses:**

There are still some inconsistencies in the use of mathematical symbols. It would be great to make the symbols defined in equations and the ones referred in the context the same.

**Questions:**

(1) In the definition of $Err^l_{\mathcal{SC}}$ and $Err^u_{\mathcal{SC}}$, whether the $2$ in the superscript means square or should be placed to subscript to represent L2-norm.

(2) Please make it more clear that how $S^{k_x}_x$ and  $S^{k_w}_x$ are sampled from $X$ and $W$.

(3) The current formalization of Algorithm 1 will lead to used of undefined symbols $\mathbf{De}_{x_u}$,

$\mathbf{De}_{w_u}$,

$\mathbf{En}_{x_u}$,

and $\mathbf{En}_{w_u}$. Please clarify that to make the Algorithm 1 strict.

---

> ### Author Response · Authors · 2025-11-27
>
> We appreciate your detailed feedback. We have reviewed the manuscript and our codebase and have edited to make it clearer and consistent.
>
> 1. Superscript or subscript in $Err$ terms
>
> We have double checked the implementation. All the reconstruction errors and prediction errors are represented using mean squared error. Therefore, the $2$ should be in superscripts rather than subscripts. We have fixed this throughout the manuscript into $||...||_2^2$ to make it clear and consistent.
>
> 2. How $S_x^{k_x}$ and $S_w^{k_w}$ are sampled
>
> In Equation 4, we presented our relation inference via do-calculus, where we marginalize out the spurious features $S$ from the prior Gaussian distribution to obtain their average effects on the relation predictions. To make the explanation clearer, we have expanded the explanation of Equation 4 to more strictly define ​​$S^{k_x}_x$ and $S^{k_w}_w$, clarifying that these represent the kx​-th and kw​-th samples drawn independently from the prior Gaussian distribution P(S), making the do-calculus marginalization process more explicit. We have added this explanation to the revised version.
>
> 3. Undefined symbols in Algorithm 1
>
> Thank you for mentioning this. We have corrected Algorithm 1 to explicitly show that the same encoders (Enx​, Enw​) and decoders (Dex​, Dew​) are applied to data from their respective domains, regardless of whether the data is labeled or unlabeled.

---

### Meta-Review · Area_Chair_HXTi · 2026-01-06

**Summary:**

This paper introduces CaSBRE, a causality-inspired semi-supervised learning framework for biomedical relation extraction. The model aims to disentangle causal and spurious features in biomedical interaction datasets (e.g., chemical–protein interactions and gene–disease associations) . The authors evaluate CaSBRE on standard biomedical relation extraction benchmarks and report improved generalization to unseen entities.

**Reviewer Concerns:**

Reviewers generally agreed that the paper addresses an important problem in biomedical NLP, but raised several concerns. First, the methodological novelty was considered incremental and weak in causal foundation. While the causality framing and Do-calculus integration are interesting, the algorithmic contributions over prior semi-supervised and causality-aware models are not clearly distinguished. The experimental evaluation was seen as lacking depth; baseline comparisons were not comprehensive, especially on unseen biomedical entities. Also, reviewers felt that the connection between the causal assumptions and empirical improvements was insufficiently justified, leaving open questions about how strongly the proposed disentanglement aligns with real distribution shifts in biomedical text. Although the authors provided clarifications during discussion, these responses did not fully alleviate concerns about novelty and evaluation rigor.

**Reviewer Scores:**

I think review cdEd may increase their score, but it is unlikely to change the borderline result of the paper.

---

### Decision · Program_Chairs · 2026-01-26

Reject